# Efficacy, safety and cost-effectiveness of methotrexate, adalimumab or their combination in non-infectious non-anterior uveitis: a protocol for a multicentre, randomised, parallel three arms, active-controlled, phase III open label with blinded outcome assessment study

Ana Belen Rivas,[1,2] Amanda Lopez-Picado,[1] Valentina Calamia,[3] Ester Carreño,[4] Lidia Cocho,[5] Miguel Cordero-Coma,[6] Alex Fonollosa,[7] Felix M Francisco Hernandez,[8] Angel Garcia-Aparicio,[9] Javier Garcia-Gonzalez,[10] Jose Juan Mondejar,[11] Leticia Lojo-Oliveira,[12] Llucí Martínez-Costa,[13] Santiago Munoz,[14] Diana Peiteado,[15] Jose Antonio Pinto,[16] Beatriz Rodriguez-Lozano,[17] Esperanza Pato,[18] David Diaz-Valle,[19] Elena Molina,[20] Luis Alberto Tebar,[1] Luis Rodriguez-Rodriguez ,[21] CoTHEIA Study Group

For numbered affiliations see end of article.

**Correspondence to**
Dr Luis Rodriguez-Rodriguez;
lrrodriguez@salud.madrid.org

## ABSTRACT

**Introduction** Non-infectious uveitis include a heterogeneous group of sight-threatening and incapacitating conditions. Their correct management sometimes requires the use of immunosuppressive drugs (ISDs), prescribed in monotherapy or in combination. Several observational studies showed that the use of ISDs in combination could be more effective than and as safe as their use in monotherapy. However, a direct comparison between these two treatment strategies has not been carried out yet.

**Methods and analysis** The Combination THerapy with mEthotrexate and adalImumAb for uveitis (CoTHEIA) study is a phase III, multicentre, prospective, randomised, single-blinded with masked outcome assessment, parallel three arms with 1:1:1 allocation, active-controlled, superiority study design, comparing the efficacy, safety and cost-effectiveness of methotrexate, adalimumab or their combination in non-infectious non-anterior uveitis. We aim to recruit 192 subjects. The duration of the treatment and follow-up will last up to 52 weeks, plus 70 days follow-up with no treatment. The complete and maintained resolution of the ocular inflammation will be assessed by masked evaluators (primary outcome). In addition to other secondary measurements of efficacy (quality of life, visual acuity and costs) and safety, we will identify subjects' subgroups with different treatment responses by developing prediction models based on machine learning techniques using genetic and proteomic biomarkers.

## Strengths and Limitations of this Study

► This is the first randomised controlled study designed to compare the efficacy of combination therapy versus monotherapy for the treatment of non-infectious uveitis in subjects with no previous immunosuppressive treatment.

► We have chosen quite a strict outcome (the requirement of a maintained controlled inflammation), more likely related to long-term outcomes (such as structural damage) and to patient-reported outcome measures.

► Despite the previous point, our primary efficacy outcome is still a surrogate marker: the achievement of this outcome does not have necessarily to translate in an improvement of outcomes more important for the patient, such as quality of life or disability.

► The requirement to control the inflammatory process early (by week 16) may cause an underestimation of drug efficacy, as it could take more time to control the inflammation but be associated with a similar long-term prognosis.

► The lack of masking could introduce bias, although treatment characteristics and proven effectivity in non-infectious uveitis, the duration of the trial (up to 52 weeks) and the need for biweekly subcutaneous injections of one of the drugs (adalimumab), we consider unpractical for the subject the use of placebo in the present trial.

**Ethics and dissemination** The protocol, annexes and informed consent forms were approved by the Reference Clinical Research Ethic Committee at the Hospital Clínico San Carlos (Madrid, Spain) and the Spanish Agency for Medicines and Health Products. We will elaborate a dissemination plan including production of materials adapted to several formats to communicate the clinical trial progress and findings to a broad group of stakeholders. The promoter will be the only access to the participant-level data, although it can be shared within the legal situation.

**Trial registration number** 2020-000130-18; NCT04798755.

## INTRODUCTION

Uveitides are potentially sight-threatening diseases:[1] worldwide, they represented up to 10% of causes of blindness (almost 4 million people).[2] Furthermore, in the European Union and the USA, after diabetic retinopathy, uveitides represent the second major treatable cause of blindness in those 20–65 years of age[3] (up to 10% of cases of blind registrations[3–6]). Additionally, a high percentage of patients suffer from uveitis-related complications, visual impairment[4 7 8] and a negative impact in quality-of-life (QoL).[9 10] Considering their higher prevalence in young to middle-aged adults,[11 12] uveitides cause an important economic, social and personal burden.[5 13–16]

The correct management of non-infectious uveitis (NIUs) is essential for preserving visual function and avoiding ocular and extra-ocular morbidity.[17] Although glucocorticoids (GCs) are the mainstay of treatment,[18] under certain circumstances, adding immunosuppressive drugs (ISDs) is needed to achieve a sustained control of the inflammatory process.[19] Several ISDs are used in the standard of care, such as methotrexate (MTX) and biological agents.[19–21]

Regarding MTX, its effectiveness in NIU has been assessed in two randomised clinical trials, compared with mycophenolate mofetil (MMF): Rathinam *et al*[22] observed that 69% and 47% of patients treated with MTX and MMF, respectively, achieved complete control of inflammation and daily oral GCs dosage ≤10 mg at 5 and 6 months (p=0.09). In a second trial,[23] randomising 200 patients, the percentage of subjects achieving a similar outcome was 67% and 57% for MTX and MMF, respectively (p>0.05), with similar tolerability. Regarding safety, adverse events (AEs) are generally mild and discontinuations due to serious adverse events (SAEs) are less common than for most ISDs,[24 25] translating it in higher retention rates.[26 27]

Regarding adalimumab (ADA), its effectiveness in NIU has been shown in two randomised controlled trials.[28 29] The VISUAL I study[28] compared ADA with placebo in 217 NIU patients with active uveitis despite ≥10 mg/day of systemic GCs. Based on the cumulative number of subjects with treatment failure in each visit, by week 25 (about 6 months), 31 of 107 (29%) subjects in the placebo group and 62 of 110 (56%) subjects in the ADA group had not suffered a treatment failure. By 50 weeks, the numbers were reduced to 24 (22%) and 51 (46%), respectively. Regarding safety, many of the AEs are sufficiently mild to not require discontinuation, such as injection site pain and antidrug antibodies

formation,[19 30 31] reflecting in a high retention rate.[32] Regarding SAEs, there was no association with higher risk in a recent meta-analysis,[33] compared with placebo or synthetic ISDs. However, ADA was associated with a higher risk of treatment discontinuation due to SAEs. One of the most important ADA's AEs are infections. However, a previous meta-analysis reported that the absolute risk was low (0.036% with TNF-alpha inhibitors vs 0.017% with placebo),[34] which probably does not represent a clinically important constraint on the use of these agents. Finally, ADA has not showed an association with a higher risk of malignancy.[33]

Although ISDs are usually used in monotherapy in uveitis, several observational studies have shown that in 21%–52% of NIU patients, the use of ISDs in monotherapy was unable to achieve a sustained control of the inflammatory process.[27 35–37] Furthermore, other studies have provided evidence that the combination of two or more ISDs could offer advantages in terms of effectiveness and tolerability,[38–41] in conditions such as Birdshot retinochoroidopathy,[38] serpiginous choroiditis,[39 40] Vogt-Koyanagi-Harada syndrome (VHK),[42 43] ocular Behçet's disease,[44 45] JIA associated uveitis,[46] sympathetic ophthalmia[47] and intermediate uveitis.[48] Regarding the beneficial of combining both MTX and ADA, several RCTs have provided evidence in other immune-mediated inflammatory diseases (IMIDs).[49] In NIU, this combination was tested in children with Juvenile Idiopathic Arthritis-associated Uveitis and a previous failure to MTX monotherapy.[50] The combination of MTX and ADA was more effective compared with MTX and placebo, although it was associated with a higher proportion of AEs (88% vs 83% of patients) and SAEs (22% vs 7%).

Despite all the evidence, a direct comparison between combination and monotherapy has not been tested yet, although some groups have adopted the use of combination therapy as the initial ISD treatment for particular conditions.[51]

There are currently no tools able to predict the response or non-response to ISDs in those NIUs needing immunosuppression, as there is marked interpersonal variation in their efficacy and toxicity. Response to the first-line ISD treatment could be an important predictor of long-term outcomes, as the continuous or repeated eye inflammation could increase the risk of structural permanent damage, leading to blindness, disability and deterioration in the QoL. Therefore, starting on the right ISD is likely a key factor in achieving a better and more cost-effective therapy, and improving the optimal allocation of healthcare resources. To achieve these aims, objectively measured and evaluated characteristics (biomarkers[52]) are required, in addition to the identification of clinical features associated with the outcomes of interest.

## METHODS AND ANALYSIS
### Study overview

The Combination THerapy with mEthotrexate and adalImumAb for uveitis (CoTHEIA) study is a phase III, multicentre, prospective, randomised, single-blinded with masked outcome assessment, parallel three arms

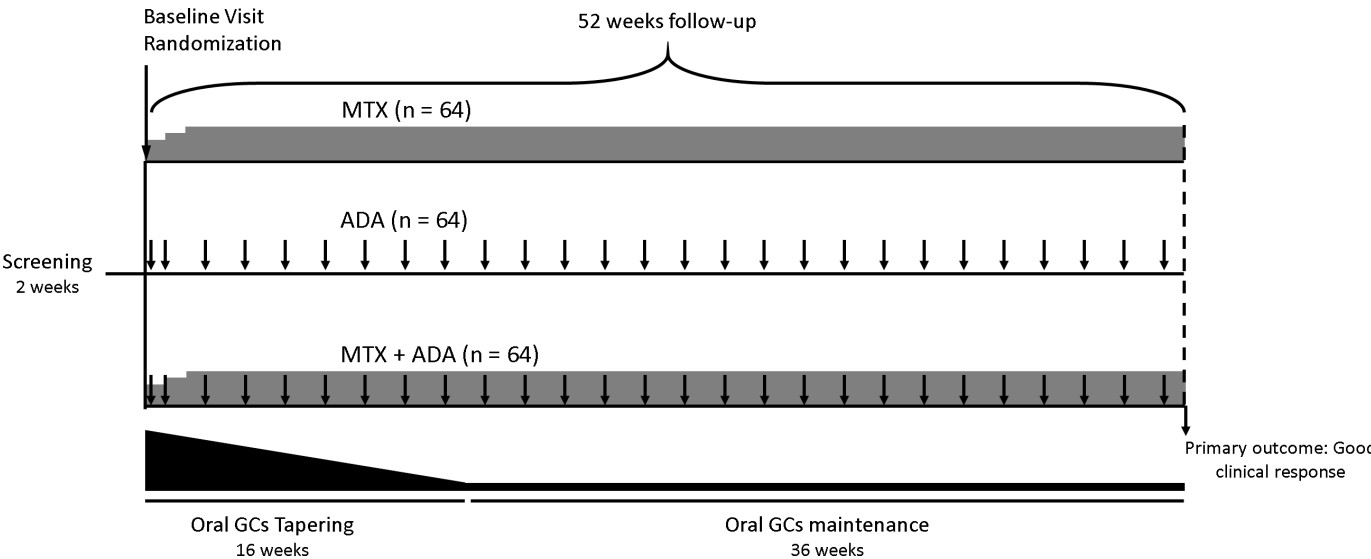

**Figure 1** Overview of the study timeline and interventions. ADA, adalimumab; MTX, methotrexate.

with 1:1:1 allocation, active-controlled, superiority study design, comparing the efficacy, safety and cost-effectiveness of MTX, ADA or their combination in non-infectious non-anterior uveitis. The duration of the treatment and follow-up will last up to 52 weeks. A 70-day follow-up clinic visit or phone call will take place to assess safety after the last study drug dose. Figure 1 provides an overview of the study timeline and interventions.

### Objectives and outcome variables

The trial main hypothesis is that the use of combination therapy with MTX and ADA will be more effective in inducing and maintaining ocular inflammatory inactivity than either drug given in monotherapy.

The *primary efficacy objective* is to establish which treatment strategy results in a higher proportion of subjects achieving a complete and maintained resolution of the ocular inflammation, on an intent to treat basis. The *primary efficacy outcome* will be the proportion of patients achieving a *Good Clinical Response* between the combination therapy arm and the single ISD arms. This outcome is defined as a complete resolution of the ocular inflammatory signs (including active chorioretinal lesions, active retinal vascular inflammation, uveitis macular oedema, presence of anterior chamber cells and presence of vitreous haze), achieved within the first 16 weeks of the study and maintained during follow-up until the end of the study (week 52); furthermore, there must not be a treatment failure due to safety or intolerability; the subject must adhere to the initial (up to week 16) oral GCs tapering protocol; all study visits from baseline to 16 weeks must be completed and at the final visit (week 52), the subject must be treated with up to 7.5 mg/day of oral prednisone (or equivalent) and up to two times a day of prednisolone acetate 1% (or equivalent). The *secondary efficacy objectives* and outcomes variables can be found at table 1.

Security-related objective will establish which treatment strategy has better tolerability. Safety outcomes will be collected in the form of AEs, physical examination and laboratory tests throughout the treatment period and up to 70 days after the last dose in this study. Gender of the subjects will be taken into account.[53]

*Pharmacogenetic and proteomic-related objectives* include identifying groups of subjects more likely to respond to the different treatment strategies, using genetic and proteomic biomarkers. For the former, subjects will be genotyped for known and validated genetics single nucleotide polymorphisms (SNPs) associated with a MTX[54] and ADA[55] response in different IMIDs. For the latter, a Discovery (shotgun proteomic analysis), Verification (targeted proteomics using multiple reaction monitoring (MRM) test) and Validation (antibody-based microarrays absolute quantification tests) phases will be carried out, following optimised protocols and procedures.[56–61]

Finally, a *Biobank substudy-related objective* will be carried out with the aim of boosting the translational research in the field of NIUs, by creating a collection of blood-derived samples (serum, plasma, total blood RNA and DNA) from the participant subjects in order to advance in the identification and validation of biomarkers associated with treatment response or deepen in the patho-physiology of these conditions.

### Settings and eligibility

The study population encompasses subjects diagnosed with non-infectious intermediate, posterior or panuveitis with active disease within 180 days before the start of the study (baseline visit), and either a documented failure to systemic or local GCs, or a chronic disease requiring GC-sparing ISD treatment. Main inclusion and exclusion criteria can be found in table 2.

Subjects will be recruited from 16 Spanish academic hospitals located in different regions of Spain (Galicia,

**Table 1** Efficacy-related secondary objectives and outcome variables

| Objective | Outcome variable |
|---|---|
| To compare the fraction of subjects who achieve a complete inflammatory ocular inactivity by week 16 | Complete abrogation of the ocular inflammatory signs, which is achieved within the first 16 weeks of the study, no treatment failure due to safety or intolerability; compliance with the initial (up to week 16) oral GCs tapering protocol, and completion of all study visits from baseline to 16 weeks |
| To compare several Patient-Reported Outcomes Measures (health-related and vision-related quality of life, anxiety and depression) between treatment strategies | EuroQuol 5D-5L<br>Visual Functioning Questionnaire-25<br>Hospital Anxiety and Depression Scale |
| To compare the presence of the clinical components of the Good Clinical Response variable between treatment strategies during follow-up | Active chorioretinal lesions; active retinal vascular inflammation; macular oedema; ACC; vitreous haze and loss of CVA secondary to inflammation at baseline, week 16, week 52 and relapse visit |
| To compare the time to relapse after week 16 between treatment strategies | The time to inflammatory relapse between groups, defined as the time from visit 16 weeks until end of the study, loss of follow-up or appearance of at least one ocular inflammatory manifestation, in those individuals achieving a Good Clinical Response by visit 16 week |
| To compare the evolution of visual acuity during follow-up | Best-corrected visual acuity (BCVA) during follow-up |
| To compare the development of anti-adalimumab antibodies during the follow-up, between those subjects treated with monotherapy and combination therapy | Assessment of anti-ADA antibodies (AAA) at week 15, 27, 51 and relapse |
| To assess the cost-utility and cost-effectiveness from both a Health System and a Societal perspective of the combination therapy and the ADA monotherapy compared with MTX given alone | Direct and indirect cost, and incremental cost effectiveness ratios. Drug costs will be calculated individually for each patient taking as reference the price published in 2022 by the Health Ministry for the Spanish Health System. Outpatient and inpatient care, other medical cost, home care and productivity loss will be estimated based on the data from eSalud database[68] and from the Minimum Basic Data Set of the Spanish Health Ministry.[69] |

ADA, adalimumab; GC, glucocorticoid; MTX, methotrexate.

País Vasco, Castilla y León, Comunidad de Madrid, Castilla La Mancha, Comunidad Valenciana and Comunidad Canaria). In addition, the Instituto de Investigación Biomédica de A Coruña (INIBIC) will carry out the proteomic analysis, and the Instituto de Investigación Sanitaria San Carlos (IdISSC) Musculoskeletal Pathology Group will be responsible for the pharmacogenetic analysis and development of prediction models for drug response.

### Recruitment

It will extend over 18 months. The participating study sites attends several 10s of new patients every year, with several hundreds or thousands being followed-up, and many of these sites act as reference centres for other secondary and tertiary hospitals. Candidate subjects will be identified from patients lists AND/OR research databases AND among the new patients attended at the sites.

Subject will not receive any financial compensation for participating in the study.

To ensure subject's retention during follow-up, we will request contact details, carry out reminder phone calls/send emails, review barriers to attend appointments, and educate them in the significance of research follow-up even if they decide to discontinue the study drugs.

### Intervention

All subjects entering the study will be centrally randomised at the baseline visit into one of three study arms, in a 1:1:1 ratio:

*Arm 1* will receive at the baseline visit ADA 80 mg SC loading dose followed a week later by 40 mg every-other-week starting at week 1. They will also receive MTX oral at the baseline visit, with initial dose of 15 mg/week, increasing up to 25 mg/week.

*Arm 2*: MTX with the same schedule as in arm 1.

*Arm 3*: ADA with the same schedule as in arm 1.

In addition to the study drugs, topical adjunctive eye medication will be allowed according to standard practice (intraocular pressure-lowering medication, cycloplegic agents, artificial tears, topical non-steroidal anti-inflammatory drugs…). The use of any biologic therapies with a potential therapeutic impact in NIUs, live vaccines, any other ISD besides the study drugs, intraocular GCs implants or intraocular surgery will be prohibited for the duration of the trial.

**Table 2** Study participant inclusion and exclusion criteria

| Inclusion criteria | Exclusion criteria |
|---|---|
| Adult subjects (≥18 years old) | Subjects with ocular histoplasmosis syndrome, or ocular masquerade syndromes |
| Diagnosed with non-infectious intermediate, posterior or panuveitis in at least one eye | Evidence or history of malignancy |
| Who have active disease within the previous 180 days before baseline, and either | Corneal, lens or vitreous opacities precluding the visualisation of the fundus |
| A documented failure to systemic or local GCs in the previous 6 months, OR | Uncontrolled intraocular pressure |
| A chronic disease necessitating GC-sparing immunosuppressive treatment (such as multifocal choroiditis with panuveitis, serpiginous choroidopathy, birdshot retinochoroidopathy, diffuse retinal vasculitis, Vogt-Koyanagi-Harada with bullous serous retinal and/or choroidal detachments or sympathetic ophthalmia) | Best-corrected visual acuity (CVA) <20/400 |
| Able and willing to self-administer subcutaneous (SC) injections or have a qualified person available to administer SC injections | History, symptoms and/or MRI findings suggestive of a demyelinating disease |
| A negative PPD test (or equivalent) and a chest X-ray (CXR) at Screening OR if positive PPD test (or equivalent) and/or a CXR consistent with prior tuberculosis (TB) exposure, the subject must initiate, be currently receiving or have documented completion of a course of TB prophylaxis therapy, according to clinical practice | History of moderate to severe congestive heart failure (NYHA class III or IV) |
|  | Behçet's disease or suspected of Behçet's disease |
|  | Previous exposure to TNFi therapies |
|  | Previous exposure to synthetic ISDs in the previous 6 months before baseline |
|  | Prior intolerability, safety issues or ineffectiveness of MTX and/or ADA |
|  | Use of GCs implants (Iluvien within 3 years, Ozurdex within 6 months before baseline) |
|  | Use of intraocular or periocular GCs injection within 90 days before baseline |
|  | Ocular surgery within 30 days before baseline |
|  | Planned (elective) eye surgery in the following 52 weeks from baseline |
|  | Proliferative or severe non-proliferative diabetic retinopathy |
|  | Neovascular/wet age-related macular degeneration |
|  | Chronic structural damage considered by the Investigator to interfere with measurement of macular thickness, impede the potential for its normalisation or can cause damage independent of the inflammatory process |
|  | Systemic inflammatory disease considered by the Investigator as likely to require high GCs dosage or prohibited medications |
|  | Presence of chronic recurring infections (HBV, syphilis), active TB and/or a history of invasive infection |
|  | Positive pregnancy test |
|  | Breast-feeding or considering becoming pregnant during the study |

ADA, adalimumab; GC, glucocorticoids; ISDs, immunosuppressive drugs; MTX, methotrexate.

To ensure a balance of patients across treatment groups and uveitis anatomic locations, patients will be stratified according to the main site of ocular inflammation (intermediate, OR posterior/panuveitis). Randomisation will not be stratified by site due to the small-expected number of subjects per site.

Block randomisation will ensure that an equal number of patients are randomised to each study arm.

The allocation sequence will consist of a computer-generated random number list generated and held in the Clinical Research Unit of Hospital Clínico San Carlos, hidden from participating Investigators. The allocation sequence will be computer-generated, and it will be implemented through the electronic case report form (eCRF; REDCap), which will assign the treatment group. Study data will be collected and managed using REDCap electronic data capture tools hosted at The Health Research Institute of the Hospital Clínico San Carlos.[62 63] REDCap (Research Electronic Data Capture) is a secure, web-based software platform designed to support data capture for research studies, providing (1) an intuitive interface for validated data capture; (2) audit trails for tracking data

**Table 3** Study activities

| Activity | Screening | Baseline | w1 | w4 | w8 | w12 | w16 | w20 | w24 | w28 | w32 | w36 | w40 | w44 | w48 | FET | Relapse | Unsch. | 70-day follow-up |
|---|---|---|---|---|---|---|---|---|---|---|---|---|---|---|---|---|---|---|---|
| Informed consent | X | | | | | | | | | | | | | | | | | | |
| Inclusion/exclusion criteria | X | X | | | | | | | | | | | | | | | | | |
| Medical/surgical history | X | X | | | | | | | | | | | | | | | | | |
| History of tobacco and alcohol use | X | X | | | | | | | | | | | | | | | | | |
| Uveitis history | X | X | | | | | | | | | | | | | | | | | |
| Vital signs/weight/height | X | X | | | | | | | | | | | | | | | X | (X) | |
| Physical exam | X | X | | | | | | | | | | | | | | | X | (X) | |
| Symptom directed physical exam | | | X | X | X | X | X | X | X | X | X | X | X | X | X | | X | | |
| Chest X-ray | X | | | | | | | | | | | | | | | X* | | | |
| Cerebral MRI† | (X) | | | | | | | | | | | | | | | | | | |
| TB screening | X | | | | | | | | | | | | | | | | | | |
| Serum pregnancy test†† | X | X | X | X | X | X | X | X | X | X | X | X | X | X | X | X | | | |
| Hepatitis B/C screening | X | | | | | | | | | | | | | | | | | | |
| Syphilis testing (FTA) | X | | | | | | | | | | | | | | | | | | |
| Randomisation | | X | | | | | | | | | | | | | | | | | |
| Best-corrected visual acuity testing | (X)‡ | (X)‡ | | X | X | X | X | X | X | X | X | X | X | X | X | X | X | | |
| Tonometry | (X)‡ | (X)‡ | | X | X | X | X | X | X | X | X | X | X | X | X | X | X | | |
| Slit lamp exam | (X)‡ | (X)‡ | | X | X | X | X | X | X | X | X | X | X | X | X | X | X | | |
| Dilated indirect ophthalmoscopy | (X)‡ | (X)‡ | | X | X | X | X | X | X | X | X | X | X | X | X | X | X | | |
| Fundus photography | (X)‡ | (X)‡ | | | | | X | | | | | | | | | X | X | | |
| Optical coherence tomography | (X)‡ | (X)‡ | | X | X | X | X | X | X | X | X | X | X | X | X | X | X | | |
| EuroQol-5D | (X)‡ | (X)‡ | | X | X | X | X | X | X | X | X | X | X | X | X | X | X | | |
| Visual Functioning Questionnaire-25 | (X)‡ | (X)‡ | | | | | X | | | | | | | | | X | X | | |
| Hospital Anxiety and Depression Scale | (X)‡ | (X)‡ | | | | | X | | | | | | | | | X | X | | |

Continued

**Table 3** Continued

| Activity | Screening | Baseline | w1 | w4 | w8 | w12 | w16 | w20 | w24 | w28 | w32 | w36 | w40 | w44 | w48 | FET | Relapse | Unsch. | 70-day follow-up |
|---|---|---|---|---|---|---|---|---|---|---|---|---|---|---|---|---|---|---|---|
| Haematology/chemistry | X | (X)§ | X | X | X | X | X | X | X | X | X | X | X | X | X | X | (X) | | |
| Monitor adverse events | | (X) | X | X | X | X | X | X | X | X | X | X | X | X | X | X | X | X | X |
| Monitor concomitant medication | X | X | X | X | X | X | X | X | X | X | X | X | X | X | X | X | X | X | X |
| Review of patient diary | | | X | X | X | X | X | X | X | X | X | X | X | X | X | X | (X) | | |
| Perform drug accountability | | | X | X | X | X | X | X | X | X | X | X | X | X | X | X | (X) | | |
| Assessment of direct and indirect costs | | | X | X | X | X | X | X | X | X | X | X | X | X | X | X | (X) | | |
| Dispense study drugs | | X | X | X | X | X | X | X | X | X | X | X | X | X | X | X | (X) | | |
| Proteomic blood sample | (X)¶ | (X)¶ | | | | | X | | | | | | | | | X | X | | |
| Pharmacogenetic blood sample | (X)¶ | (X)¶ | | | | | | | | | | | | | | | | | |
| Adalimumab levels and antiadalimumab antibodies | (X)¶ | (X)¶ | | | | | X** | | | X** | | | | | | X** | X** | | |

*Chest X-ray will be performed at the 52 weeks/early termination visit in case the patient had a positive TB test at baseline.

†Cerebral MRI will be performed only in those subjects with intermediate uveitis or signs of intermediate uveitis.

‡Values from the baseline visit or from a screening visit up to 14 days before the baseline visit will be used as the values from the baseline visit.

§Total blood count, renal and liver function test from up to 4 weeks before the baseline visit will the required before entering the study.

¶Blood sample will be drawn after signature of the informed consent, as soon as possible, ideally before start of oral GCs.

**Blood samples for the antiadalimumab antibodies will be drawn the week before visits 16, 28, 52. Early termination and relapse visits, up to 24 hours before the patient is scheduled to receive the ADA dosage, and always before the medication is taken.

††Only in women in childbearing

FET, final (w52)/early termination visit; ISD, Immunosuppressive Drug; NYHA, New York Heart Association; PPD, Purified protein derivative ; TNF, Tumor Necrosis Factor; Unsch, unscheduled visit; w, week.

**Table 4** Planned methods of statistical analysis for efficacy-related secondary outcomes

| Outcome variable | Statistical analysis |
|---|---|
| Complete abrogation of the ocular inflammatory signs, which is achieved within the first 16 weeks of the study, no treatment failure due to safety or intolerability; compliance with the initial (up to week 16) oral GCs tapering protocol and completion of all study visits from baseline to 16 weeks | MHT, stratified by NIU location. If p-value<0.05, pairwise comparisons using MHT stratified by NIU location will be carried out with Bonferroni adjustment of the pairwise p-values |
| EQ5D | GEE models nested by patient[70 71] and adjusted by study visit (continuous) and treatment arm (discrete) will be carried out, using a Gaussian family and Identity as link function. Different covariable structures will be tested (independent and exchangeable) and compared using the Bayesian Information Criteria. Time x study arm interactions will assess different effects of time in the evolution of the outcome by arm. P-value<0.05 will be considered as a significant interaction |
| Visual Functioning Questionnaire-25 Hospital Anxiety and Depression Scale | Changes between Baseline and w16, and the FET visit will be compared between treatment groups using ANOVA |
| Active chorioretinal lesions; active retinal vascular lesions; macular oedema; ACC; vitreous haze and loss of CVA secondary to inflammation at baseline, week 16, week 52 and relapse visit | MHT, stratified by NIU location. If p-value<0.05, pairwise comparisons using MHT stratified by NIU location will be carried out with Bonferroni adjustment of the pairwise p-values |
| The time to inflammatory relapse between groups, defined as the time from visit 16 weeks until end of the study, loss of follow-up or appearance of at least one ocular inflammatory manifestation, in those individuals achieving a Good Clinical Response by visit 16 week | Time to relapse between arms will be analysed using log-rank test at a two-sided significance level of 5%. Dropouts due to reasons other than inability to maintain a Good Clinical Response will be considered as censored observations at the time of dropping out. Only subjects able to achieve a Good Clinical Response by visit week 16 will be analysed. We will consider both the time until the onset of the first inflammatory manifestation, each inflammatory manifestation, the first inflammatory manifestation that does not resolve in the following 4 weeks and each inflammatory manifestation that does not resolve in the following 4 weeks. |
| Best-corrected visual acuity during follow-up | GEE models |
| Assessment of anti-ADA antibodies (AAA) at weeks 15, 27, 51 and relapse | MHT, stratified by NIU location. If p-value<0.05, pairwise comparisons using MHT stratified by NIU location will be carried out with Bonferroni adjustment of the pairwise p-values |
| Direct and indirect cost, and incremental cost effectiveness ratios | Cost utility and cost-effectiveness analysis from the perspectives of the National Health System and the Society will be performed. EQ5D scores will derive utility values representing health related quality of life. QALYs will be calculated by the area under the curve assuming a linear evolution of EQ5D values between visits. The average number of QALYs per patient will be calculated for each study arms. Effectiveness will be defined using our primary efficacy outcome. An average cost per patient will be calculated for each study arm, including direct (drug cost, outpatient and inpatient care and other medical cost) and indirect cost (home care, productivity loss): Incremental cost-effectiveness ratios will be calculated, using the MTX monotherapy arm as comparison |

ACC, Anterior Chamber Cells; ADA, adalimumab; ANOVA, analysis of variance; CVA, corrected visual activity; EQ5D, EuroQuol 5D-5L; FET, final (w52)/early termination visit; GCs, glucocorticoids; GEE, generalised estimating equations; MHT, Mantel-Haetzel; MTX, methotrexate; NIU, non-infectious uveitis; QALYs, quality-adjusted life years.

manipulation and export procedures; (3) automated export procedures for seamless data downloads to common statistical packages and (4) procedures for data integration and interoperability with external sources. After obtaining the informed consent for enrolment and confirming that all eligibility requirements have been met, the Unmasked Investigator (see next section) will log into the eCRF and perform the randomisation. Unmasked investigators will then give their assigned treatment to the subjects.

### Authorised rescue medication

During whole study up to two inflammatory relapses (unilateral or bilateral) will be allowed, one during the first period (baseline–week 16: ACC relapse) and one during the second period (week 16–week 52: any location), before declaring the lack of response to the assigned medication. Based on the location and severity of the relapse, a protocolised rescue treatment with Topical AND/OR Local OR Oral GCs will be allowed. In case inflammation cannot be suppressed in 4 weeks, or a new relapse takes place after 4 weeks, it will be declared treatment failure and the subject will exit the study.

### Masking

For the duration of the trial, both the study subject and the Unmasked Investigators will be aware of the treatment

| Table 5 | Trial registration data and protocol summary |
|---|---|
| **Data category** | **Information** |
| Primary registry and trial identifying number | EudraCT: 2020-000130-18 |
| Date of registration in primary registry | 9 March 2021 |
| Secondary identifying numbers | ClinicalTrials.gov: NCT04798755 |
| Source of monetary or material support | Instituto de Salud Carlos III |
| Primary sponsor | Fundación para la Investigación Biomédica del Hospital Clínico San Carlos |
| Contact for queries | Luis Rodriguez-Rodriguez, MD (lrrodriguez@salud.madrid.org) |
| Public title | Combination THerapy with mEthotrexate and adalImumAb for uveitis (CoTHEIA) |
| Scientific title | Efficacy, safety and cost-effectiveness of methotrexate, adalimumab or their combination in non-infectious non-anterior uveitis: a multicentre, randomised, parallel three arms, active-controlled, phase III open label with blinded outcome assessment study |
| Countries of recruitment | Spain |
| Health condition or problem studied | Non-infectious non anterior uveitis |
| Intervention(s) | Intervention 1: Adalimumab 40 mg every-other-week (plus a 80 mg SC loading dose)+methotrexate oral, up to 25 mg/week, both for a duration of 52 weeks |
| | Intervention 2: Methotrexate with the same schedule as in intervention 1 |
| | Intervention 3: Adalimumab with the same schedule as in intervention 1 |
| Key inclusion and exclusion criteria | Age≥18 years old Diagnosed with non-infectious intermediate, posterior or panuveitis in at least one eye Active ocular disease Lack of satisfactory response to systemic and/or local glucocorticoid (GC) therapy AND/OR diagnoses of a chronic disease usually necessitating GC-sparing immunosuppressive treatment |
| Study type | Phase III, multicentre, prospective, randomised, single-blinded with masked outcome assessment, parallel three arms with 1:1:1 allocation, active-controlled, superiority study design |
| Date of first enrolment | N/A |
| Target sample size | 64 per treatment arm (192 in total) |
| Recruitment status | Not yet recruiting |
| Primary outcome | Complete resolution of the ocular inflammatory signs, which is achieved within the first 16 weeks of the study, and maintained during follow-up until the end of the study (week 52) |
| Key secondary outcomes | Safety, cost-effectiveness |

N/A, not applicable.

assigned. Considering the differences between study drugs (MTX and ADA) regarding appearance, route of administration and schedule, in order to make them unaware of the medication prescribed, the use of placebo would be necessary. However, taking into account that both drugs have been proven effective in the treatment of NIU, the duration of the trial (up to 52 weeks), and the need for biweekly subcutaneous (SC) injections of one of the drugs (ADA), we consider unethical for the subject the use of placebo in the present trial.

Besides the Unmasked Investigators, the rest of participating investigators will be considered Masked Investigators (ophthalmologists performing the clinical eye exams, visual acuity examiners, Optical Coherence Tomography (OCT) operators, fundus photographers, fundus graders and administrators of subject's questionnaires) and will not be aware of the treatment assigned to prevent bias in study outcomes.

Several steps will be taken to avoid the Masked Investigators discovering the subject assignment: they will have no part in handling or prescribing medication; subjects will be given dark bags to place and keep their medication in throughout the trial and study visits to minimise the chances of the Masked Investigators seeing the medications. Additionally, subjects will meet with the Unmasked Investigator first, before seeing any Masked Investigators, keeping any study medication in his/her office for the entire patient visit; reviewing the appearance of any AE with the subjects before seeing any Masked Investigators, and reminding the subjects not to discuss their dosing and mediation name with the Masked Investigators.

## Study procedures

Study visits will be the baseline visit, visits at weeks 1, 4, and every 4 weeks thereafter until (a) the subject is determined as unable to achieve complete resolution of the ocular inflammatory process by week 16, OR; (b) the subject is determined as unable to maintain a complete resolution of the ocular inflammatory process, between weeks 16 and 52 OR; (c) the subject completes 52 weeks of this clinical trial, OR; (d) the study is stopped due to the findings of the Data Security Monitoring Board, OR and (e) the subject meets any of the study finalisation criteria.

Informed consent has to be acquired before carrying out any study procedure, including screening tests (online supplemental file 1 contains a sample informed consent). Before baseline (when the subject is randomised), several screening visits can take place up to 14 days before that visit, in order to obtain all the required complementary tests to assess eligibility. The visit window for all scheduled visits is ±3 days through week 4 and ±7 days for all visits following the week 4 study visit. Table 3 provides an overview of the study activities.

After the trial, all patients will return to standard care and will be able to continue with their assigned study medication.

## Data management and monitoring

An Unmasked Investigator in each centre will review and crosscheck for consistency and completeness all data collected in RedCap within 24 hours of the study visit. If the forms are not filled out completely, the responsible person will be contacted for providing the missing data. An external monitoring service will conduct regular checks of the data regarding errors and inconsistencies, supervising data collection, management and quality control, and will submit queries to the site investigators.

## Safety

Non-serious AEs will be defined as an unfavourable and unintended sign (including abnormal laboratory findings), symptom or disease temporally associated with the use of the study medication or procedure, whether or not considered related to the study medication. Both MTX and ADA may be temporarily suspended in case of non-serious AEs, such as laboratory alterations, infections, and intolerance. In addition, MTX dosage may also be reduced in case of laboratory alterations and/or intolerance. Dose reductions and temporary discontinuations will be decided by the Masked Investigator, who must remain unaware of the medication and dosage assigned to the subject, so preconceptions regarding the study drugs do not interfere with their management. The Masked Investigator will issue three recommendations; one for each medication arm the subject may be included. Then, the Unmasked Investigator will implement the recommendation according to the arm the subject is included.

SAEs will be defined as any AE that results in death, is life threatening, requires hospitalisation, or prolongation of existing hospitalisation, results in persistent or significant disability or incapacity, or is a congenital anomaly or birth defect.

All AEs will be recorded in clinical records and a medically qualified investigator will assess the relationship of SAEs to the study medications.

All SAE that occurs during the trial must be reported immediately by mail or fax to the Pharmacovigilance Unit within 24 hours of its occurrence. The investigator will complete and sign the SAE notification form to be sent by e-mail.

The Pharmacovigilance Unit will review the form received and, if applicable, ask for additional information to the investigator. The investigator should provide the requested information or any new information regarding the case, especially if the initial assessment in severity or causality has been changed, following the procedure previously described.

The Pharmacovigilance Unit is responsible for submitting as soon as possible all Suspected Unexpected Serious Adverse Reactions (SUSARs) collected during the study to the Spanish Health Authorities and Ethic Committee, according with the Spanish legislation: no later than 15 calendar days (seven in case of fatal or life-threatening cases) after first knowledge by the sponsor that the case meets the minimum criteria for expedited reporting.

## Data and Safety Monitoring Committee

Only after the Data and Safety Monitoring Committee (DSMC) reviews and approves the protocol will patients be enrolled. In addition, the group will meet regularly throughout the study and review information on data quality, enrolment, patient retention and study outcomes according to DSMC charter. The DSMC will be independent from the Sponsor, Funding Body and Principal Investigator and will include experts in the fields of ophthalmology, rheumatology and epidemiology.

## Adherence

Adherence will be monitored using a Patient's Diary (where they will register all study medication administered outside of the study visit (ie, at home), including reasons for missing dosages) and by verifying the returned empty medication (partially/completely empty blisters of study medication, AND/OR cartons and sharps containers for MTX, ADA and oral GCs).

## Patient and public involvement

No patient involved.

## Statistical analysis plan

The intention to treat (ITT) set will include all subjects who were randomised.

The safety set will consist of all subjects who received at least one dose of study medications. Per protocol analysis will also carry out. Missing data will be imputed using multiple imputation-chained equations.

### Sample size determination

*ADA arm*: after 1 year (50 weeks) of treatment, 20.3% of NIU patients achieved an outcome similar to our *primary efficacy outcome*.[28]

*MTX arm*: after 6 months of treatment, 65% of NIU patients achieve also similar outcome;[64] we estimate that of those patients, 18%[65] and 16%[66] will be unable to maintain our outcome due to AEs and inefficacy, respectively; therefore, at 1 year, 43% will achieve our *primary efficacy outcome*. This figure will be assumed as the percentage of subjects treated with monotherapy achieving the primary efficacy outcome.

*MTX+ADA arm*: we will assume that combination therapy will increase the percentage of subjects achieving our *primary efficacy outcome* by 23% compared with the monotherapy arms.

To detect statistically significant differences between groups with a power of 80% and a significance level of 0.05, it will be necessary to recruit 54 patients per study arm (162 in total). Since the follow-up period is 52 weeks, losses of 15% will be assumed, increasing the sample size to 64 patients per study arm (192 patients in total). In order to test the difference between the treatments, superiority or relevant clinical improvement has been considered from a delta of 5% of the effect.

## Planned methods of statistical analysis for efficacy-related objectives

The primary analysis will be a Mantel–Haenszel test (MHT), stratified by NIU location, comparing the combination therapy arm and the single ISDs arms (both arms combined), and performed in the ITT set. Achievement of the Good Clinical response will be considered the binary response variable: those subjects achieving a Good Clinical versus those not achieving the outcome (regardless the cause). The exposure variables will be the treatment arm: those subjects receiving combination therapy versus those receiving either monotherapy. Uveitis location will be the strata: intermediate uveitis versus posterior OR panuveitis. If the results of the primary analysis are significant, then pairwise comparisons using MHT stratified by NIU location will be carried out. P values of the pairwise comparisons will be adjusted using the Bonferroni method. No interim efficacy-related analysis will be carried out.

Secondary efficacy-related analyses are designed to test the hypothesis that treatment assignment affects a given outcome, after controlling for selected covariates. Details can be found at table 4.

## Planned methods of statistical analysis for safety-related aims

The safety analysis will be performed in the safety set. Treatment-emergent AEs (events with an onset date on or after the first study drug administration until 70 days following the last study drug administration) will be summarised by treatment group using descriptive statistics. SAEs with onset after informed consent but before the first study drug administration will be considered as pretreatment SAEs and reported separately.

AEs will be tabulated by system organ class and preferred term whereby the most current implemented MedDRA dictionary will be used. In addition, summaries by severity and relationship to study drug will be done. Certain AEs, such as serious or severe, leading to premature withdrawal, will be listed and described in detail. AEs of special interest for treatments will be defined in the statistical analysis plan and analysed separately. In addition to the descriptive statistics provided, Fisher's exact test will be used for comparisons between treatment groups.

## Genetic analysis

SNPs genotypes will be determined by real-time PCR amplification using Taqman probes and following standard procedures. Duplicate genotypes of 10% of the samples, concordance (all $p>0.05$) with the Hardy–Weinberg equilibrium and with SNP frequencies in the HapMap European collection will be used for quality control.

Comparison of the proportion of subjects achieving the primary efficacy outcome between genotypes will be carried out using a $\chi^2$ test or Fisher's exact test, when required. Dominant, recessive and additive models of effects will be considered for each SNP. P values will be adjusted using the Bonferroni method. Each study arm will be analysed separately.

## Proteomic analyses

Discovery phase: Shotgun proteomic analysis[56–58] will be performed on serum samples from a representative group of subjects from each study arm with extreme responses (n=20): those achieving the *primary efficacy outcome* and those not being able to achieve a good clinical response by week 16.

Verification phase: targeted proteomics will be used for verification of protein markers with predictive potential in a randomly selected larger samples set (n=80). After MRM tests, relative quantification methods of the proteins will be designed.[59]

Validation phase: best candidates from the previous phase will be validated using absolute quantification tests (antibody-based microarrays) in the whole set of subjects.[60 61]

## Patient subgroup identification

Based on baseline visits patient's characteristics (demographic, disease and clinical-related variables), those genotypes significantly associated with the primary efficacy outcome, and the previously identified serum proteins in the verification phase, prediction models for MTX, ADA and Combination therapy response will be developed using a machine learning method (Random Forests[67]). Models' performance will be assessed with the area under the receiver operating characteristic curve, and calibration curves. Models using only clinical data, only biomarkers data and the combination of both will be developed, to assess the contribution of biomarkers to the models' predictive ability. Due to the modest sample size we plan to recruit, we will not divide our sample in training, validation and test data sets. All subjects will be considered as part of the training data set, and a 10-fold cross-validation will be carried out to internally validate our models.

## ETHICS AND DISSEMINATION

The protocol (version 2, 11 September 2020), annexes and informed consent forms have been approved by the Clinical Research Ethic Committee (CREC) at the Hospital Clínico San Carlos (Madrid, Spain) and the Spanish Agency for Medicines and Health Products (AEMPS). The local approvals corresponding to the participating centres will be obtained and documented before starting the study in that centre as per centre requirements. The promoter will be the CREC interlocutor corresponding to his/her centre in everything related to the present study. It will keep CREC informed of the evolution of the study in the centre and of the possible minor incidents and modifications that may occur. Any relevant modification to the protocol must receive express approval from the reference CREC and the AEMPS before its implementation, unless there are risk circumstances for the participating subjects, in which case the precise measures to

ensure the integrity of the study subjects will be implemented immediately, pending the corresponding approvals. The trial is registered at clinicaltrialsregister.eu (EudraCT:2020-000130-18) and clinicaltrials.gov (NCT04798755). This study involves human participants and was approved by Hospital Clínico San Carlos Ethics Committee, approval ID '20/510–EC_M'. Participants gave informed consent to participate in the study before taking part.

No study related activities will be carried out before obtaining a written informed consent from the patient. The investigator will be responsible for: (a) providing each patient with an information sheet about the trial and the objectives, methods, foreseeable benefits and potential risks of the study, (b) discussing the information with the patient, in terms understandable for the subject and (c) explaining to patients that they are totally free to refuse their participation in the study or to abandon it at any time and for any reason. If the subject agrees to participate in the Biobank Substudy, a second independent informed consent will be collected, which will include the possibility of storing the samples not used in the present study in the Collection of Samples for Research in Rheumatic Diseases of the Rheumatology Department of the Hospital Clinico San Carlos (and, in a second phase, when the Coordinating Investigator of this study deems it appropriate and always in the event that the samples has not been used up, the remainder will be stored in the Hospital Clinico San Carlos Biobank).

All the data will be treated confidentially at any times: data will be pseudonimised, the paper forms will be kept in locked cabinets, the eCRF is located in a secure server and the person in charge of the analysis will not be able to access identification data of the subjects.

Table 5 summarises the study protocol and trial registration information.

Data obtained through this study may be provided to qualified researchers with academic interest in uveitis. Data or samples shared will be coded, and donated to a Registered Biobank and made available under legal requirement. Approval of the request and execution of all applicable agreements are prerequisites to the sharing of data with the requesting party.

Regarding dissemination, in order to communicate the clinical trial progress and findings to a broad group of stakeholders, we will elaborate a dissemination plan which will include production of materials adapted to scientific meetings, scientific publications, patients and other stakeholders. A summary of the final version of the study protocol will be made available through the Spanish Clinical Trial Registry and Clinicaltrials.gov database. The promoter will be the only with access to the participant-level data, following the regulation on data protection.

**Author affiliations**

[1]Unidad de Investigación Clinica y Ensayos Clínicos, Hospital Clínico San Carlos, IdISSC, Madrid, Spain

[2]Departamento de Enfermería. Facultad Enfermería, Fisioterapia y Podología, Universidad Complutense de Madrid, Madrid, Spain

[3]Unidad de Proteómica. Grupo de Investigación de Reumatología (GIR), Instituto de Investigación Biomédica de A Coruña, Complexo Hospitalario Universitario de A Coruña, and Universidade da Coruña, A Coruna, Galicia, Spain

[4]Department of Ophthalmology, University Hospital Fundacion Jimenez Diaz, and University Hospital Rey Juan Carlos, Madrid, Madrid, Spain

[5]Department of Ophthalmology, IOBA (Institute of Applied OphthalmoBiology), University of Valladolid, and Hospital Clínico Universitario de Valladolid, Valladolid, Castilla y León, Spain

[6]Uveitis Unit, University Hospital of León, IBIOMED, and University of León, Leon, Spain

[7]Department of Ophthalmology, BioCruces Bizkaia Health Research Institute, Cruces University Hospital, University of the Basque Country, Barakaldo, País Vasco, Spain

[8]Department of Rheumatology, Hospital Universitario de Gran Canaria Dr Negrin, Las Palmas de Gran Canaria, Spain

[9]Department of Rheumatology, Hospital Universitario de Toledo, Toledo, Spain

[10]Department of Rheumatology, Hospital Universitario 12 de Octubre, Madrid, Comunidad de Madrid, Spain

[11]Department of Ophthalmology, Hospital General Universitario de Alicante, Alicante, Comunidad Valenciana, Spain

[12]Section of Rheumatology, Hospital Universitario Infanta Leonor, Madrid, Spain

[13]Department of Ophthalmology, Hospital Universitario Doctor Peset, Valencia, Spain

[14]Department of Rheumatology, Hospital Universitario Infanta Sofia, San Sebastian de los Reyes, Madrid, Spain

[15]Department of Rheumatology, Hospital Universitario La Paz, Madrid, Madrid, Spain

[16]Department of Rheumatology, Complexo Hospitalario Universitario de A Coruña, A Coruna, Galicia, Spain

[17]Department of Rheumatology, Hospital Universitario de Canarias, Santa Cruz de Tenerife, Canarias, Spain

[18]Department of Rheumatology, Hospital Clínico San Carlos, IdISSC, Madrid, Madrid, Spain

[19]Department of Ophthalmology, Hospital Clínico San Carlos, IdISSC, Madrid, Spain

[20]Department of Pathology, Hospital Clínico San Carlos, IdISSC, Madrid, Spain

[21]Musculoskeletal Pathology Group, Fundacion para la Investigacion Biomedica del Hospital Clinico San Carlos, IdISSC, Madrid, Spain

**Collaborators** CoTHEIA Study GroupMara Albert Fort (Hospital Universitario Doctor Peset), Mayte Ariño Gutierrez (Hospital Clínico San Carlos), Pedro Arriola Villalobos (Hospital Clínico San Carlos), Joseba Artaraz Beobide (Hospital Universitario Cruces), Antonio Atanes Sandoval (Complejo Hospitalario Universitario A Coruña), Jose Manuel Benitez Del Castillo Sanchez (Hospital Clínico San Carlos), Francisco Blázquez (Instituto Universitario de Oftalmología Aplicada, Universidad de Valladolid), Pablo Borges Deniz (Hospital Universitario Fundación Jiménez Díaz), Lara Borrego Sanz (Hospital Clínico San Carlos), Gabriela Bustamante Sanchez-Arnedo (Instituto Universitario de Oftalmología Aplicada, Universidad de Valladolid), Bruno Casco Silva (Complejo Hospitalario Universitario de Toledo), Ricardo Cuiña Sardiña (Hospital Clínico San Carlos), Almudena De Pablo Cabrera (Hospital Universitario 12 de Octubre), Maria Del Mar Esteban Ortega (Hospital Universitario Infanta Sofía), Patricia Fernandez Puente (Instituto de Investigación Biomédica de A Coruña), Dalifer Dayanira Freites Nuñez (Hospital Clínico San Carlos), Javier Garcia Bella (Hospital Clínico San Carlos), Sara Garcia Carazo (Hospital Universitario La Paz), Luis Garcia Onrubia (Instituto Universitario de Oftalmología Aplicada, Universidad de Valladolid), Jose Maria Garcia Ruiz De Morales (Complejo Asistencial Universitario de León), Jose Antonio Gegundez Fernandez (Hospital Clínico San Carlos), Vanesa Hernandez Hernandez (Complejo Hospitalario Universitario de Canarias), Inés Hernanz Rodriguez (Hospital Universitario Fundación Jiménez Díaz), Jose Maria Herreras Cantalapiedra (Instituto Universitario de Oftalmología Aplicada, Universidad de Valladolid), Ventura Hidalgo Barrero (Hospital Universitario La Paz), Maria De La Vega Jovani Casano (Hospital General Universitario de Alicante), Maria Isabel Lopez Rodriguez (Complejo Hospitalario Universitario A Coruña), Sara López Sierra (Hospital Universitario Doctor Peset), Virginia Lozano Lopez (Complejo Hospitalario Universitario de Canarias), Alfredo Madrid Garcia (Hospital Clínico San Carlos), Rosalia Mendez Fernandez (Hospital Clínico San Carlos), Pilar Nozal Aranda (Hospital Universitario La Paz), Eugenio Perez Blazquez (Hospital Universitario 12 de Octubre), Chamaida Plasencia Rodriguez (Hospital Universitario La Paz), Sheila Recuero Diaz (Hospital Universitario Fundación Jiménez Díaz), Ignacio Robles Barrena (Hospital Universitario Rey Juan Carlos), Mª Esther Rodriguez Almaraz (Hospital Universitario 12 de Octubre), Fayna Maria Rodriguez Gonzalez (Hospital Universitario de Gran Canaria Doctor Negrín), Guadalupe Rodriguez Martinez (Complejo Hospitalario Universitario A Coruña), Fredeswinda Isabel Romero Bueno (Hospital Universitario Fundación Jiménez Díaz), Ioana Ruiz Arruza

(Hospital Universitario Cruces), Beatriz Sanchez Marugan (Hospital Universitario Infanta Leonor), Olga Sanchez Pernaute (Hospital Universitario Fundación Jiménez Díaz), Armelle Schlinker Giraud (Hospital Universitario La Paz), Alvaro Seijas Lopez (Complejo Hospitalario Universitario A Coruña), Patricia Simon Alonso (Complejo Hospitalario Universitario A Coruña), Marta Tejera Santana (Hospital Universitario de Gran Canaria Doctor Negrín), Elia Valls Pascual (Hospital Universitario Doctor Peset).

**Contributors** L-RR, AL-P, EC and DD-V conceived of the study. L-RR, AL-P, EP, DD-V, MC-C, AF, AG-A, JG-G, SM, ABR and LAT initiated the study design and helped with implementation. L-RR is the funding holder. AL-P and ABR provided statistical expertise in clinical trial design. VC provided expertise regarding the proteomic analysis design. L-RR provided expertise regarding the pharmacogenetic analysis. AL-P, ABR and L-RR will conduct the primary statistical analysis. ABR, AL-P, VC, EC, LC, MC-C, AF, FMFH, AG-A, JG-G, JJM, LL-O, LM-C, SM, DP, JAP, BR-L, EP, DD-V, EM, LAT and L-RR contributed to refinement of the study protocol and approved the final manuscript.

**Funding** This work was supported by the Instituto de Salud Carlos III, grant number [ICI19/00020]. Sponsor: Fundación para la Investigacion Biomédica del Hospital Clínico San Carlos. Executive Committee: Administrative and executive arm of the clinical trial, providing overall oversight for the study and making decisions on day-to-day operational issues (Study Coordinator (Luis Rodriguez-Rodriguez), a representative from the Spanish Clinical Trial Network (Amanda López Picado), and 5 Site Directors (these seats will be rotatory, with changes every 6 months months)); Data Coordinating and Analysis Committee: Supervising data collection,management and quality control, designing the statistical analysis plan, performing unmasked data analysis and preparing interim and final reports for the Data Security Monitoring Board and the Executy Committee (Study Coordinator (Luis Rodriguez-Rodriguez), a representative from the Spanish Clinical Trial Network (Amanda López Picado) and Ester Carreño); Biobank and Biomarker Identification Committee (Maintaining an up-to-date manual of operations for blood extraction, processing and storage, and monitoring procedures adherence, supervising biological sample collection, sample shipment coordination, coordinating the phamacogenetic and proteomic analysis (Study Coordinator (Luis Rodriguez-Rodriguez), a representative from the Instituto de Salud Carlos III Biobank Platform (Elena Molino), a representative the Instituto de Investigación Biomédica de A Coruña, a representative from, the Data Coordinating and Analysis Committee); Data Security Monitoring Committee (PierGiorgio Neri, Andrew Dick, Loreto Carmona).

**Disclaimer** The study sponsor had no role in the study design; collection, management, analysis and interpretation of data; writing of the report and the decision to submit the report for publication.

**Competing interests** None declared.

**Patient consent for publication** Not required.

**Provenance and peer review** Not commissioned; externally peer reviewed.

**ORCID iD**
Luis Rodriguez-Rodriguez http://orcid.org/0000-0002-2869-7861

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
