## [Reviewer comments · BMJ Open]

ARTICLE DETAILS

TITLE (PROVISIONAL)	Efficacy, safety and cost-effectiveness of methotrexate, adalimumab, or their combination in non-infectious non-anterior uveitis: a protocol for a multicenter, randomized, parallel 3 arms, active-controlled, phase 3 open label with blinded outcome assessment study
AUTHORS	Rodriguez-Rodriguez, Luis; Rivas, Ana; Lopez-Picado, Amanda; Calamia, Valentina; Carreño, Ester; Cocho, Lidia; Cordero-Coma, Miguel; Fonollosa, Alex; Francisco Hernandez, Felix; Garcia-Aparicio, Angel; Garcia-Gonzalez, Javier; Mondejar, Jose Juan; Lojo-Oliveira, Leticia; Martínez-Costa, Lluçí; Munoz, Santiago; Peiteado, Diana; Pinto, Jose Antonio; Rodriguez-Lozano, Beatriz; Pato, Esperanza; Diaz-Valle, David; Molina, Elena; Tebar, Luis

VERSION 1 – REVIEW

REVIEWER	Wen, Zehuai Guangdong Provincial Hospital of Chinese Medicine, Key Unit of Methodology in Clinical Research
REVIEW RETURNED	28-May-2021

GENERAL COMMENTS	This is a well-designed randomized trial which wants to compare methotrexate, adalimumab or their combination for non-infectious non-anterior uveitis. The manuscript complied well with the requirements of the SPIRIT 2013 Statement. My comment is following. Authors mentioned that this is a superiority study design, but the superiority comparison and the superiority margin are not explained in the sample size estimation section.
---

REVIEWER	Oh , Baek Lok Seoul National University Hospital
REVIEW RETURNED	26-Aug-2021

GENERAL COMMENTS	The authors proposed a valuable phase III trial protocol comparing the efficacy, safety, and cost-effectiveness of MTX, ADA, or their combination in NINAU. Although the study design seems to be scientific and reasonable, I have two issues to be addressed. #1. Is there any rescue treatment for the patient with sight-threatening severe inflammation during the study? #2.
--

	What is the justification of "70-day follow-up" for the assessment of safety after the last study drug dose? How about 60 days or 90 days?
--	--

VERSION 1 – AUTHOR RESPONSE

Reviewer: #1

Dr. Zehuai Wen, Guangdong Provincial Hospital of Chinese Medicine

Comments to the Author:

This is a well-designed randomized trial which wants to compare methotrexate, adalimumab or their combination for non-infectious non-anterior uveitis. The manuscript complied well with the requirements of the SPIRIT 2013 Statement. My comment is following.

1. Authors mentioned that this is a superiority study design, but the superiority comparison and the superiority margin are not explained in the sample size estimation section.

Response: We apologize for not having included that important piece of information. As mentioned in the manuscript, and based on previous evidence, we estimate that in the control group (i.e. monotherapy) the primary efficacy outcome will be achieved by 43% of the subjects. We estimate that 23% more of the subjects (i.e. 66%) treated with combine therapy will achieve the same outcome. The text from the manuscript has been modified accordingly (page 23, in yellow)

Reviewer: #2

Dr. Baek Lok Oh , Seoul National University Hospital

Comments to the Author:

The authors proposed a valuable phase III trial protocol comparing the efficacy, safety, and cost-effectiveness of MTX, ADA, or their combination in NINAU. Although the study design seems to be scientific and reasonable, I have two issues to be addressed.

#1. Is there any rescue treatment for the patient with sight-threatening severe inflammation during the study?

Response: We thank the reviewer for raising this question, and we apologize for not having included this information in the manuscript. During follow-up, in the first 16 weeks of the trial, up to one anterior chamber relapse will be allowed, which will be treated using a protocolized topical corticosteroids schedule. From week 16 to 52, up to one relapse will be allowed, managed with a protocolized topical and/or oral corticosteroids schedule based on the location and severity of the manifestations.

Answering your question, yes, there is rescue treatment in case of ocular inflammation, including sight-threatening manifestations. We have included a new subsection in the Methods Section to reflect this information (page 15, in blue).

#2. What is the justification of "70-day follow-up" for the assessment of safety after the last study drug dose? How about 60 days or 90 days?

Response: We apologize for the the lack of details in this regard. The follow-up of the study has been calculated considering safety issues after knowing the treatment that every subject has received.

VERSION 2 – REVIEW

REVIEWER	Wen, Zehuai Guangdong Provincial Hospital of Chinese Medicine, Key Unit of Methodology in Clinical Research
REVIEW RETURNED	06-Dec-2021

GENERAL COMMENTS	This is a well-designed randomized controlled trial which wants to assess the efficacy, safety and cost-effectiveness of the combination of methotrexate and adalimumab compared to a monotherapy for non-infectious nonanterior uveitis. The manuscript reported basically following the SPIRIT statement. I only have some minor comments. 1. For the cost-effectiveness assessment, it suggested reporting how to estimate the direct and indirect cost in the three arms.2. Authors mentioned that “hypothesis is that the use of combination therapy with MTX and ADA will be more effective than either drug given in monotherapy”. As for the sample size calculation, does it need to consider superiority or non-inferiority tests?3. Please check the spelling error such as “percentaje”.
---

VERSION 2 – AUTHOR RESPONSE

Reviewer #1

This is a well-designed randomized controlled trial that wants to assess the efficacy, safety and cost-effectiveness of the combination of methotrexate and adalimumab compared to a monotherapy for non-infectious non-anterior uveitis. The manuscript reported basically following the SPIRIT statement. I only have some minor comments.

1. For the cost-effectiveness assessment, it suggested reporting how to estimate the direct and indirect cost in the three arms.

Response: We apologize for not having included that information in the manuscript. We have included this information in Table 1: “Drug costs will be calculated individually for each patient taking as reference the price published in 2022 by Health Ministry for Spanish Health System. Outpatient and inpatient care, other medical cost, home care and productivity loss will be estimated based on the data from eSalud database and from the MBDS of the Spanish Health Ministry” (page 11).

2. Authors mentioned that “hypothesis is that the use of combination therapy with MTX and ADA will be more effective than either drug given in monotherapy”. As for the sample size calculation, does it need to consider superiority or non-inferiority tests?

Response: We apologize the misunderstanding, the principal aim of this clinical trial is to test the differences among groups referring as “superiority” when there is a clinical improvement obtained with the treatment; In this case, it is considered relevant more than 5% (delta 0.05). This information has been added to the new version of the manuscript (Page 23, 3rd paragraph).

3. Please check the spelling error such as “percentaje”.

Response: We apologize for not having corrected yet those typos. Manuscript has been reviewed and corrected.

We also want to point out that we have modified the informed consent, and therefore we have uploaded the new approved version as supplementary file 1.

In addition, we have included “Behçet’s disease or suspicion of Behçet’s disease” as a new exclusion criteria, as advised in the first meeting of the Data Safety Monitoring Board.

VERSION 3 – REVIEW

REVIEWER	Wen, Zehuai Guangdong Provincial Hospital of Chinese Medicine, Key Unit of Methodology in Clinical Research
REVIEW RETURNED	09-Feb-2022
GENERAL COMMENTS	There are no more reviews left.